# Can Weight Regularization in the Last Layer Reduce Dimensional Collapse?

## Abstract

The dimensional collapse of representations in self-supervised learning is an ever-present issue. One notable technique to prevent such collapse of representations is using a multi-layered perceptron network called Projector. In several works, the projector has been found to heavily influence the quality of representations learned in a self-supervised pre-training task. However, the question still lingers. *What role does the projector play?* If it does prevent the collapse of representations, then why doesn't the last layer of the encoder take up the role of projector in the absence of an MLP one? In this work, we intend to study what happens inside the projector by examining the rank dynamics of the same and the encoder through empirical study and analysis. Through mathematical analysis, we observe that the effect of rank reduction predominantly occurs in the last layer. Furthermore, we show that applying weight regularization only in the last layer yields better performance than when used on the whole network (WeRank), both with and without a projector. Empirical results justify that our interpretation of the role of the projector is correct.

## 1 Introduction

Self-supervised learning aims to learn representations without any human annotations. Recent works like SimCLR (Chen et al., 2020a), MoCov2 (Chen et al., 2020b), DCL (Yeh et al., 2022), BYOL (Grill et al., 2020), Barlow Twins (Zbontar et al., 2021), etc. present frameworks which allow learning of representations which are similar for semantically similar samples. However, this objective may lead to a complete collapse of representations when the representations of all samples get mapped trivially to a single point in the representation space.

Various techniques, such as using a large batch size (Chen et al., 2020a), momentum encoder (Chen et al., 2020b; Grill et al., 2020), stop gradient (Chen & He, 2020), feature whitening (Bardes et al., 2022; Zbontar et al., 2021) and clustering (Caron et al., 2020), have been used to prevent the complete collapse of representations. However, contrastive self-supervised learning still suffers from dimensional collapse, where the embedding vectors only span a lower-dimensional subspace. Dimensional collapse occurs when the variance of information along some dimensions becomes insignificant. We avoid saying that variance will be zero because information content along any dimension can never be entirely zero in practical terms.

In Hua et al. (2021), the author discusses that dimensional collapse is mainly related to a strong correlation between information flowing through different dimensions. This challenging issue of dimensional collapse has also been addressed in works like Balestriero & LeCun (2022), RankMe (Garrido et al., 2023), DirectCLR (Jing et al., 2022) and WeRank (Pasand et al., 2024). These works also stress the importance of full-rank representations for better performance on downstream tasks. However, WeRank does not provide any mathematical insight into the dimensional collapse of representation. In DirectCLR, the attempt at investigating the causes of the dimensional collapse is limited to toy examples, and only uses a truncated vector for training, leaving the last few dimensions non-trainable. This, however, is not fully capable of preventing dimensional collapse. We instead use the full output vector for both training and evaluation as well. Furthermore, we show that, unlike WeRank (Pasand et al., 2024), it is not necessary to apply the weight

regularisation on the whole network, thereby reducing the computation overhead from $\mathcal{O}(L \cdot n^3)$ to $\mathcal{O}(n^3)$, where $L$ is the scalar factor that comes naturally as shown in the later section Sec. 3.6.

Even though dimensional collapse still occurs even when a projector is used, we understand that the projector plays an important role in reducing dimensional collapse. The role of the projector has been studied previously in works like Gupta et al. (2022); Song et al. (2023); Xue et al. (2024). However, none of the above works explores *how* the use of projectors diminishes the effect of dimensional collapse.

In this work, we first empirically verify the decorrelating effect of InfoNCE loss. Next, we try to determine what happens inside the projector and its role in self-supervised contrastive learning. We further investigate the phenomenon occurring in the encoder layer that causes degradation in performance in the case of dimensional collapse resulting from negligible variance along some feature dimensions, both with or without a projector. Finally, we employ a simple strategy for self-supervised learning, both with and without using a projector, which verifies our mathematical conclusion. We summarize our contributions as follows:

- We investigate the role of the projector in self-supervised contrastive learning in the light of dimensional collapse. To our knowledge, this is the first work to do so.
- We further investigate the phenomenon in the encoder when not using a projector for contrastive self-supervised pre-training, giving us more insight into the phenomenon of dimensional collapse.
- Based on our findings, we propose a simple strategy to improve performance in contrastive self-supervised pre-training by applying weight regularization only on the last layer.

## 2  Related Works

SSL methods take different approaches to prevent a complete collapse of representations. Instance discrimination methods like SimCLR (Chen et al., 2020a), MoCov2 (Chen et al., 2020b), and DCL (Yeh et al., 2022) use repulsion between negative samples to prevent complete collapse. However, in addition to the negative repulsion in the InfoNCE loss, they also use a projector which projects the encoder output representations into a lower-dimensional space before computing the InfoNCE loss. Methods like DeepCluster (Caron et al., 2018) and SwAV (Caron et al., 2020) use a clustering-based instance-group discrimination approach. However, dimensional collapse persists according to Garrido et al. (2023) and Jing et al. (2022).

Architectures similar to the above are also seen in dimension contrastive methods like BYOL (Grill et al., 2020), where the extra predictor for predicting the output of the projector from the momentum updated target encoder and l2-normalization prevents complete collapse. SimSiam (Chen & He, 2020), on the other hand uses a stop-gradient method to prevent the same. WMSE (Ermolov et al., 2021), ZeroCL (Zhang et al., 2022b) uses feature whitening to prevent collapse.

Non-contrastive methods like Barlow Twins (Zbontar et al., 2021) aim to decorrelate the feature dimensions to reduce redundancy in the output embeddings, thereby preventing dimensional collapse. However, Barlow Twins fails to perform without the projector, as we will see in the later subsections. VICReg (Bardes et al., 2022) uses a covariance term in the loss to do feature decorrelation like Barlow Twins. However, according to Garrido et al. (2023), even these methods are not free from dimensional collapse.

Hua et al. (2021) discusses that the strong correlation between dimensions of the representation vector is the primary cause of dimensional collapse, and uses feature decorrelation to prevent it and improve performance. Balestriero & LeCun (2022) also uses a decorrelation loss like VICReg as a method to prevent dimensional collapse and learn optimal representations. Gupta et al. (2022) shows that a projector prevents low-rank backbone features, thereby preventing dimensional collapse. However, it does not explore the reason behind it. This work primarily discusses that a learnable projection head is a way of mitigating the shortcomings of contrastive loss and helps in learning generalizable representations. A detailed discussion of the relationship between downstream performance and embedding rank is also presented in Garrido et al. (2023). WeRank (Pasand et al., 2024) uses the same feature decorrelation strategy to deduce that the weight norm of each layer should be as close to the identity matrix as possible to prevent dimensional collapse.

DirectCLR (Jing et al., 2022) achieved considerable success in preventing collapse. This work mainly proposed two findings as the possible causes of dimensional collapse: (1) implicit regularization due to

over-parametrization of networks, and (2) strong augmentations. However, in terms of performance (linear evaluation accuracy), it falls short of SimCLR with a non-linear projector.

DINO (Caron et al., 2021) introduced self-distillation without labels, where a student network learns from a momentum-updated teacher using normalized feature matching, enabling Vision Transformers to learn semantic features without supervision. iBOT (Zhou et al., 2021) extended DINO by combining self-distillation with masked image modeling, allowing simultaneous global and local representation learning through patch-level prediction. DINOv2 (Oquab et al., 2024) further refined the framework with large-scale curated data, improved regularization, and stronger transferability, yielding universal visual features competitive with supervised models. I-JEPA (Assran et al., 2023) departed from contrastive and pixel-level objectives by predicting high-level latent representations of masked regions, emphasizing abstraction and context understanding. Together, these methods progressively evolve self-supervised learning from instance discrimination toward semantically rich, transferable, and predictive representations.

Recent work such as VCReg (Zhu et al., 2023) extends the idea of weight and feature regularization by encouraging high-variance and low-covariance representations to enhance feature diversity and prevent neural collapse. This approach aligns conceptually with our method and WeRank (Pasand et al., 2024), as both aim to improve representation quality and transferability through more balanced and decorrelated feature learning.

## 3 Methodology

### 3.1 Notations

In this subsection, we discuss the notations followed in the rest of the paper for mathematical derivations and discussions.

Table 1: Table for Notations

| Symbols | What it means |
|---------|---------------|
| $\mathcal{L}$ | Loss function |
| $x$ | Input |
| $f$ | Encoder |
| $h$ | output of Encoder $f$ |
| $g$ | Projector |
| $z$ | Output of Projector |
| $s_{ij}$ | Cosine similarity between the projector output embeddings of the samples of $x_i$ and $x_j$ |
| $B$ | Batch Size |
| $D$ | Number of dimensions in the encoder output embedding |
| $d_0$ | Number of trainable dimensions in the encoder output embedding |
| $d_r$ | Number of non-trainable dimensions in the encoder output embedding |
| $W_l$ | Weight matrix of $l$-th layer with dimensions $D_o \times D_i$ |
| $D_o$ | Output dimensions |
| $D_i$ | Input dimensions |
| $W_l^i$ | $i$-th row of the weight matrix $W_l$ |
| $\mathcal{I}$ | Mutual information |

### 3.2 Preliminaries

In this work, we consider SSL pre-training with SimCLR as the baseline. Let us denote $f$ and $g$ as the encoder and the projector, respectively. The encoder output and the projector output embeddings are denoted by $h = f(x)$ and $z = g(f(x))$, respectively, where $x$ denotes the input sample. The total number of dimensions in the encoder output embedding is given by $D$. $d_0$ and $d_r$ denote the number of dimensions of the encoder output embedding, which are trainable and non-trainable or fixed to a constant value. To learn the representations, the InfoNCE loss is given by,

$$\mathcal{L}_{infonce} = -\mathbb{E}_i \left[ \frac{exp(s_{ii+})}{exp(s_{ii+}) + \sum_{\substack{j=1 \\ j \neq i}}^{B} exp(s_{ij})} \right] \tag{1}$$

where $s_{ii+}$ and $s_{ij}$ are the cosine similarity between the projector output embeddings of the samples of positive pair $(x_i, x_{i+})$ obtained by augmentations applied on the sample $x_i$, and the samples of negative pair $(x_i, x_j)$, respectively, and $B$ denotes the batch size.

In the later subsections, we divide the output embeddings into 2 parts, which we refer to as trainable and non-trainable dimensions. We define the trainable part of an embedding to consist of those dimensions through which the gradient propagation is allowed to flow. At the same time, the non-trainable part of the embedding means the opposite.

### 3.3 Definitions

**(D1) Dimensional Collapse:** Dimensional Collapse occurs when the embedding vectors span a lower-dimensional subspace $\mathbb{R}^{D'}$ instead of the entire embedding space $\mathbb{R}^D$, where $D' < D$.

**(D2) Information Bottleneck:** The information bottleneck (IB) is a principle for representation learning that aims to extract a compressed representation of a variable that is as predictive as possible of a target variable. The IB framework is based on finding a compressed representation, $T$, of an input variable, $X$, that preserves the maximum relevant information about a target variable, $Y$. This is expressed through a constrained optimization problem. The goal is to find the representation $T$ that maximizes the relevance $\mathcal{I}(T; Y)$ for a given compression level $\mathcal{I}(T; X)$. This can be expressed as a Lagrangian optimization problem: $\min_{p(t|x)} \mathcal{I}(T; X) - \beta \mathcal{I}(T; Y)$, where $\beta$ is the Lagrangian multiplier.

**(D3) High-level representations:** In deep learning, high-level representations are the abstract, complex features learned by the deeper layers of a neural network. This is because deep networks process information hierarchically, transforming simple inputs into progressively more complex and meaningful features. The difference between low-level and high-level representations is best understood by following the flow of information through a deep network; that is, the low-level representations are simple, local patterns detected by the initial, shallower layers of the network, whereas high-level representations are obtained by combining the simpler features from shallower layers.

**Relevance to Downstream Tasks**: The principle of information bottleneck is utilised in the downstream task to discard irrelevant information while retaining useful information related to the downstream task. The high-level representations, that is, the task-specific representations in the deeper layers, are also relevant to the downstream task. Finally, the dimensional collapse, which can occur for both the self-supervised pre-training and supervised training stages, causes the learning to occur in a lower-dimensional subspace rather than in the high-dimensional embedding space. A larger utilisation of the learning subspace results in better performance in the downstream task.

### 3.4 Motivation

In this work, the main motivation is to study the phenomenon occurring inside the projector in the self-supervised contrastive learning scenario and what happens in the absence of it. In DirectCLR (Jing et al., 2022), it is stated that in instance discrimination-based contrastive learning, even though the presence of positive and negative samples should prevent the dimensional collapse of representations intuitively, it still occurs.

We find this to be true empirically as shown in Fig. 1a, where we observe that the magnitudes of the sorted eigenvalue spectrum dip considerably when the encoder is trained without a non-linear projector than when trained with one. Similar findings are also reported in Gupta et al. (2022). Furthermore, methods using feature decorrelation to prevent dimensional collapse, like Balestriero & LeCun (2022) or Hua et al. (2021),

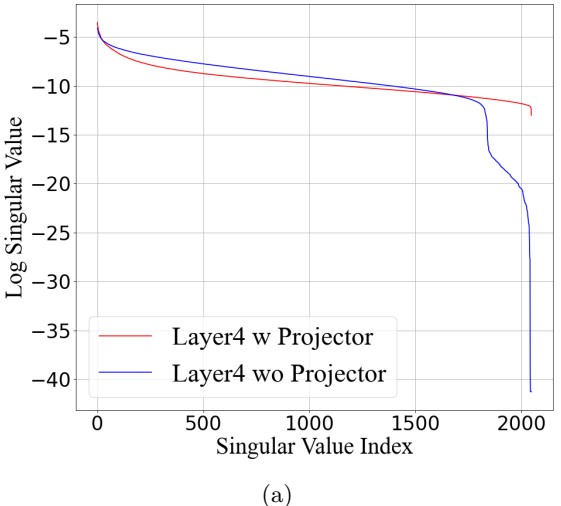 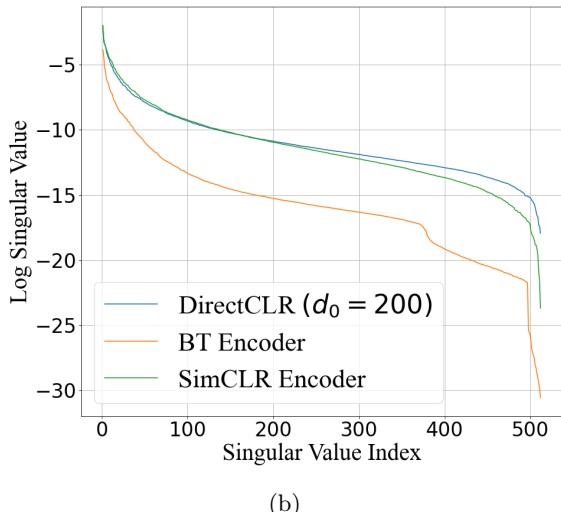

(a)             (b)

Figure 1: (a) Singular value plots of the covariance matrix of ResNet50 encoder output embeddings pre-trained on ImageNet100 using SimCLR with and without a projector. 'Layer4' indicates the last layer in the ResNet50 encoder. 'blue': without (wo) projector, 'red':with projector. (b) Singular value plots of Barlow Twins and SimCLR encoders compared with DirectCLR. The plot (a) exhibits that without the projector, the singular values of the covariance matrix drop sharply. A similar observation is also found in Barlow Twins (b), while the vanilla SimCLR and DirectCLR method prevents the sharp decline.

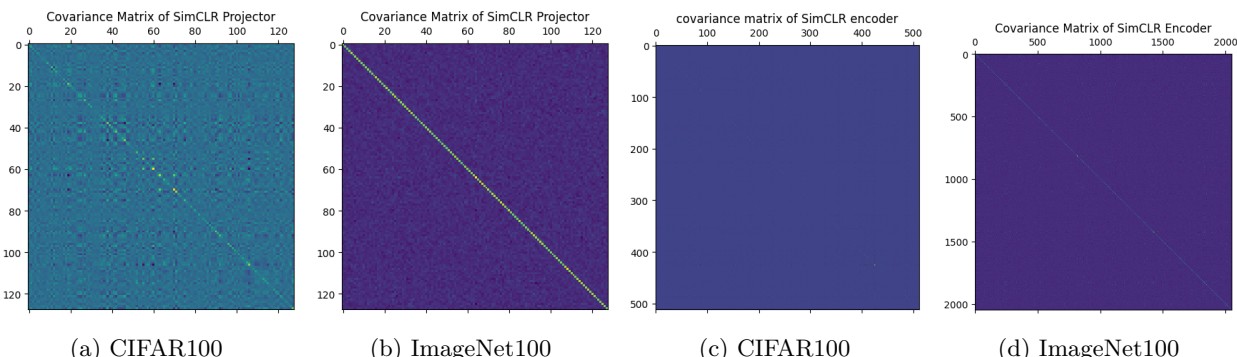

(a) CIFAR100      (b) ImageNet100      (c) CIFAR100      (d) ImageNet100

Figure 2: Covariance matrices of output embedding of the projector for SimCLR trained on (a) CIFAR100 and (b) Imagenet100. Covariance matrices of embeddings from the SimCLR encoder trained on (c) CIFAR100 and (d) ImageNet100. On both CIFAR100 and ImageNet100, we can observe the projector output embedding exhibiting low covariance in the off-diagonal components, which points towards the decorrelation effects of the InfoNCE loss (a and b). Furthermore, the decorrelation effect is propagated partially to the encoder output embeddings, as evident from the uniform nature of the covariance matrix values (c and d). Best viewed at 300%.

still suffer from dimensional collapse. This is primarily due to the low-rank embeddings of shallower layers, that is, from the encoder (Pasand et al., 2024). To determine the role of the projector, we empirically study whether the InfoNCE loss has a decorrelating effect. Then we try to analyze the dynamics of the projector through rank decomposition of the covariance matrix and how it prevents dimensional collapse.

### 3.5 Does InfoNCE have a decorrelating effect?

According to Zhang et al. (2022a), InfoNCE also acts as a decorrelating loss, similar to Barlow Twins (Zbontar et al., 2021) or Balestriero & LeCun (2022). In Fig. 2a and 2b, we show the covariance matrix of the output feature dimensions. From the covariance matrix of the embeddings of the CIFAR100 and ImageNet100

datasets, we can see that the magnitudes of the diagonal elements of the covariance matrix are much higher than the non-diagonal ones. This shows that the InfoNCE loss has a decorrelating effect, as shown in Zhang et al. (2022a). However, from Fig. 2c and 2d, we see that the diagonal nature of the covariance matrix of the encoder output embeddings is not present. This proves that even if the loss enforces feature decorrelation on the projector output embeddings, it is possible to obtain low-rank output embeddings from the encoder.

### 3.6 Understanding the events in Projector in case of Dimensional Collapse

It is empirically observed in DirectCLR (Jing et al., 2022) that InfoNCE loss fails to properly optimize the parameters of a network without a projector and results in dimensional collapse. Barlow Twins (BT) (Zbontar et al., 2021) performs better than most contrastive learning frameworks on benchmark datasets, but not when implemented without a projector, even though a decorrelation loss is applied. The singular value spectrum in Fig. 1b plots the singular values of BT (w/o projector), SimCLR (w/o projector), and DirectCLR. We can see that the dimensional collapse effect in BT is greater than in SimCLR w/o projector, even though it is trained directly using a decorrelation-based loss. According to Zhang et al. (2022a), the projector is essential for a decorrelation-based framework too, even though both InfoNCE and the loss used in Zbontar et al. (2021) have a decorrelating component. So, the question arises, ***what exactly happens after the addition of a non-linear projector towards the prevention of dimensional collapse?***.

**A Linear Algebraic perspective:** In RankMe (Garrido et al., 2023) and DirectCLR (Jing et al., 2022), the authors have shown that without a projector, the embeddings from the pre-trained encoder have a low rank. Does a low-rank embedding indicate that the useful information can be approximated using fewer dimensions? But then, ***why does it lead to worse performance, if that is the case?***. How is it different from the information bottleneck theory of the projector? (Ouyang et al., 2025).

### 3.6.1 Weight Norm Analysis

It is important to note that in DirectCLR (Jing et al., 2022), a part of the output vector $z$ is left unchanged, that is, the kernels leading to the unchanged part of $z$ still have the randomly initialized weights at the end of pre-training. Now, these randomly initialized weights have non-zero variance. However, when the rank of the encoder output embeddings is reduced, it practically means that the variance of the information along those dimensions is very low. A very low variance means there is almost no useful information available in that dimension. Thus, when not using a projector, the reduction in rank in the last layer embedding covariance matrix indicates that there is little variance of information along some of the embedding dimensions. Consequently, this indicates that the norm of the weights in the last layer $\|W_l^i\|^2 < \epsilon$, where $\epsilon$ is very small, and $Var(W_l^i) \to 0$, where $W_l^i$ is the layer weights corresponding to the $i$-th output embedding dimension of layer $l$.

**Theoretical Setup:** The theoretical framework primarily revolves around a black box neural network, which outputs embeddings, followed by two MLP layers represented by the weight matrices $W_{l-1}$ and $W_l$ or $W$ for the last layer. The term $h$ represents the output of the final layer, while $x$ represents the input to the weight matrices. Additionally, for simplicity, we consider no activation or batchnorm for the last two weight matrices.

***Proposition 1*** : The norm of the weight vectors associated with the collapsed dimensions becomes zero.

***Proof*** : We can prove the above proposition by a simple deduction. Let $x \in \mathbb{R}^{D_i}$ be the embeddings with dimensions $N \times D_i$, and $W \in \mathbb{R}^{D_o \times D_i}$ be the weight matrix with dimensions $D_o \times D_i$. To consider only a single dimension $i$, we take the $i$-th row of the weight matrix as $W^i \in \mathbb{R}^{1 \times D_i}$. Let, the covariance of $x$ be defined as

$$\Sigma = Cov(x) = \mathbb{E}\left[\left(x^T - \mathbb{E}[x^T]\right)\left(x^T - \mathbb{E}[x^T]\right)^T\right] \tag{2}$$

It is to be noted that, since we are dealing with the last layer only, it is to be assumed that the input to the last layer does not have any collapsed dimension, and $\Sigma$ is positive definite with the smallest eigenvalue $\lambda_{min} > 0$.

$$h_i = W_l^i x_l^T = W_l^i \cdot \left(W_{l-1} x_{l-1}^T\right) = \left(W_l^i W_{l-1}\right) x_{l-1}^T \tag{3}$$

$$Var(h_i) = W_l^i \left(x_l^T - \mathbb{E}[x_l^T]\right) \left(x_l^T - \mathbb{E}[x_l^T]\right)^T (W_l^i)^T = W_l^i \Sigma_l \left(W_l^i\right)^T = \|W_l^i \Sigma_l^{\frac{1}{2}}\|_2^2 < \delta \tag{4}$$

Since variance cannot be exactly zero, we consider $Var(h_i) < \delta$. In that case, $\|W^i\|^2$ lies in the approximate left null space (Ito et al., 2010) of $\Sigma_l$. Now, if $\lambda_{min}$ be the minimum eigenvalue of $\Sigma_l$, then

$$\delta > Var(h_i) = W_l^i \Sigma_l \left(W_l^i\right)^T \geq \lambda_{min} \|W_l^i\|^2 \tag{5}$$

This gives us,

$$\|W_l^i\|^2 < \frac{\delta}{\lambda_{min}} < \delta' \tag{6}$$

In the general case, if we consider $Cov(h_i, h_j)$, we can deduce the following,

$$\begin{aligned} Cov(h_i, h_j) &= \left(W^j.x^T - \mathbb{E}[W^j.x^T]\right).\left(W^i.x^T - \mathbb{E}[W^i.x^T]\right)^T \\ &= \left(W^j.x^T - W^j.\mathbb{E}[x^T]\right).\left(W^i.x^T - W^i.\mathbb{E}[x^T]\right)^T \\ &= W^j \left(x^T - \mathbb{E}[x^T]\right).\left(x^T - \mathbb{E}[x^T]\right)^T .W^{iT} = W^j \Sigma_l W^{iT} \end{aligned} \tag{7}$$

This gives us,

$$|Cov(h_i, h_j)| = |W^j \Sigma_l (W^i)^T| \leq \lambda_{max} \|W^j\| \|W^i\| \tag{8}$$

For non-constant $x$ with large enough variance, $|Cov(h_i, h_j)|$ will be very small or close to zero, if $\|W^i\|$ is small, regardless of $\|W^j\|$. In this case, a row of the covariance matrix becomes small or close to zero. Hence, the rank reduction effect or dimensional collapse occurs. (Derivation of Eqn. 8 in Appendix A)

However, in the reverse case if $Var(W^i) < \epsilon'$ but $\|W^i\|^2$ is not small, we can trivially deduce that, $Var(h_i)$ or $Cov(h_i, h_j)$ is also large. Therefore, the contribution of the dimension $i$ to the rank of the covariance matrix cannot be ignored.

Equation 6 shows that, as the variance of an output dimension approaches zero, the corresponding weight-row norm must shrink proportionally. Therefore, dimensional collapse in the embedding space of the final layer is necessarily accompanied by vanishing weight norms for those output coordinates.

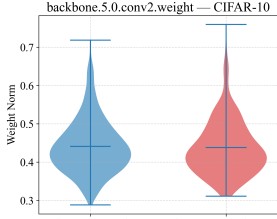 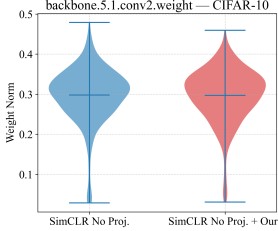 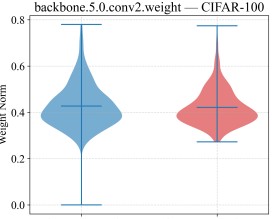 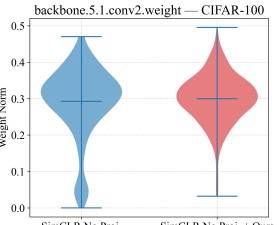

(a) CIFAR10 - Last Layer - 1st Block - 2nd Conv
(b) CIFAR10 - Last Layer - 2nd Block - 2nd Conv
(c) CIFAR100 - Last Layer - 1st Block - 2nd Conv
(d) CIFAR100 - Last Layer - 2nd Block - 2nd Conv

Figure 3: Comparison of channel-wise weight norm distribution of SimCLR without projector trained without (blue) and with (red) our proposed weight regularization in the last layer for the penultimate and final layers of the projector. The distribution of weight norms of the SimCLR without the projector shows that the weights of the second convolutional layer of the second resnet block in the last layer (h) **go very close to 0.0**. Whereas the proposed method (p) can raise the minimum **away from 0.0** despite having similar mean values. Best viewed at 200%.

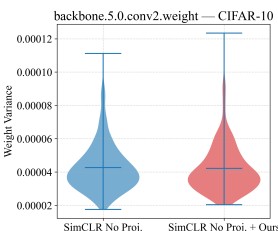 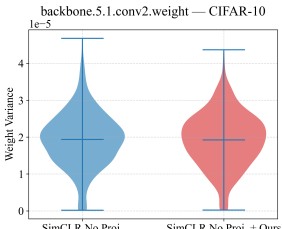 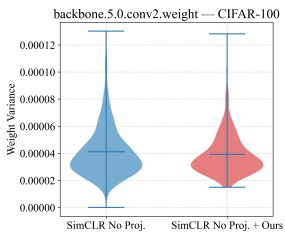 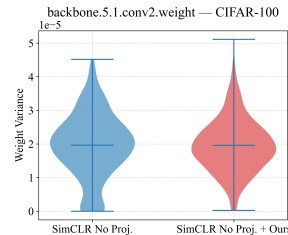

(a) CIFAR10 - Last Layer - 1st Block - 2nd Conv

(b) CIFAR10 - Last Layer - 2nd Block - 2nd Conv

(c) CIFAR100 - Last Layer - 1st Block - 2nd Conv

(d) CIFAR100 - Last Layer - 2nd Block - 2nd Conv

Figure 4: Comparison of channel-wise unbiased variance of weight of SimCLR without projector trained without (blue) and with (red) our proposed weight regularization in the last layer for the penultimate and final layers of the projector. The distribution of unbiased variances of the weight of the SimCLR without the projector shows that the weights of the second convolutional layer of the second resnet block in the last layer (h) **go very close to 0.0**. Whereas the proposed method (p) can raise the minimum **away from 0.0** despite having similar mean values. Best viewed at 200%.

**Key Takeaway 1: Weights with near-zero norm prevent learning of high-level representations**
Thus, if the norm of the weights and the variance of the weights corresponding to the collapsed dimensions become close to zero, it prevents the kernels in the deeper layers in the backbone encoder from learning class-specific high-level representations, consequently hampering the downstream performance, as the presence of lower values in the weight variances reduces their representation learning capacity.

**Empirical Justification:** We look at the distribution of the channel-wise weight norms of the convolutional layers in the penultimate and last layer of the encoder (Fig. 3), trained using the SimCLR framework, but without a projector. We can observe that there are several samples in the distributions which have values very close to zero, indicating the presence of collapsed dimensions. Additionally, we also present the effect of applying a weight regularization in the last layer only when trained without a projector in the same figure.

The importance of these empirical findings lies in the fact that they directly corroborate the "Key Takeaway 1" mentioned previously. By definition, the last layer corresponds to learning the high-level and more complex representations. Having a negligible weight norm and a negligible variance reduces the representation learning capacity of the layers in concern. From Fig. 3 and 5b, we can combine the empirical results to infer that in the absence of a projector, both the covariance of the embedding dimensions and the weight norm become negligible along a few dimensions. Hence, considering that the concerned weights are from the last layer, we can safely state that the kernels in the deeper layers are incapable of learning useful information in the absence of a projector.

### 3.6.2 Propagation of Low-Rank Representations

In the case of a perfect decorrelating effect of the InfoNCE loss as discussed in the previous subsection, the flow of information would be maximized, resulting in better semantic representation learning and consequently better downstream performance. *But the decorrelation effect of the InfoNCE on the encoder output embeddings does not maximize the flow of information through the encoder output dimensions.* A similar approach to prevent dimensional collapse was also presented in WeRank (Pasand et al., 2024), where the authors used weight regularization by whitening the weight covariance matrix.

**Empirical justification and observations** : We study the singular value plots of the penultimate and final layer of the encoder backbone in Fig. 5. We see an increase in the number of high-valued singular values in the final layer of the backbone when using a projector. However, the eigenvalue spectrum is almost unchanged with or without a projector in the penultimate layer of the ResNet backbone encoder. *Thus, we can say that the effect of rank reduction is observed only on the final layer, when not using a separate*

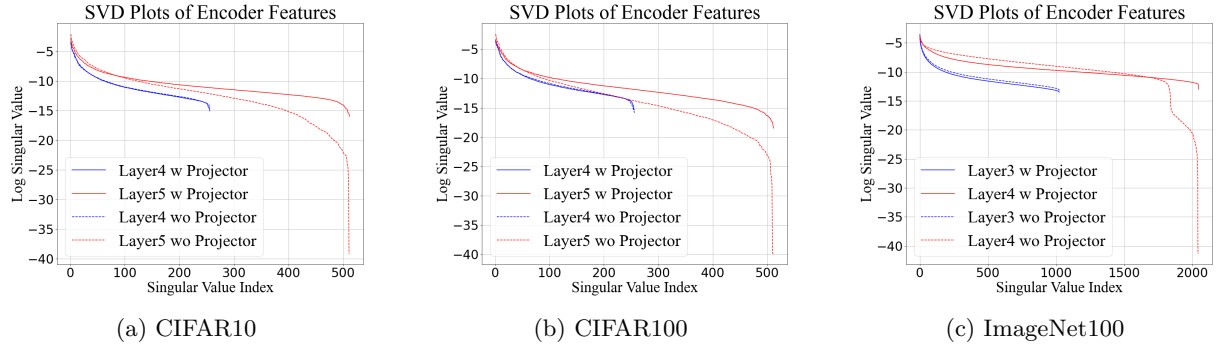

(a) CIFAR10          (b) CIFAR100          (c) ImageNet100

Figure 5: Singular values plots of the penultimate and final layers of the encoder with and without the projector. The plots above show the effect of the presence and absence of the projector on the last and the penultimate layer of the encoder. The last layer (Layer5) shows a clear dimensional collapse without the projector in all three datasets, CIFAR10, CIFAR100 and ImageNet100. However, there is a negligible change in the singular value plot for the penultimate layer (Layer4), which agrees with our previous observation that the decorrelation effect is not fully propagated towards the shallower layers. Best viewed at 300%.

*projector, and the final layer of the backbone acts as the make-shift or pseudo projector.* However, according to our observation, it is wise to say that the rank reduction effect is observed on the layer from which the final embedding is taken for contrastive loss computation, while the eigenspectrum of the previous layer output embeddings shows almost no change. This is confirmed from the plots of eigenvalues in Fig. 6, where we observe that the eigenvalues for the last layer of the projector are significantly low in magnitude.

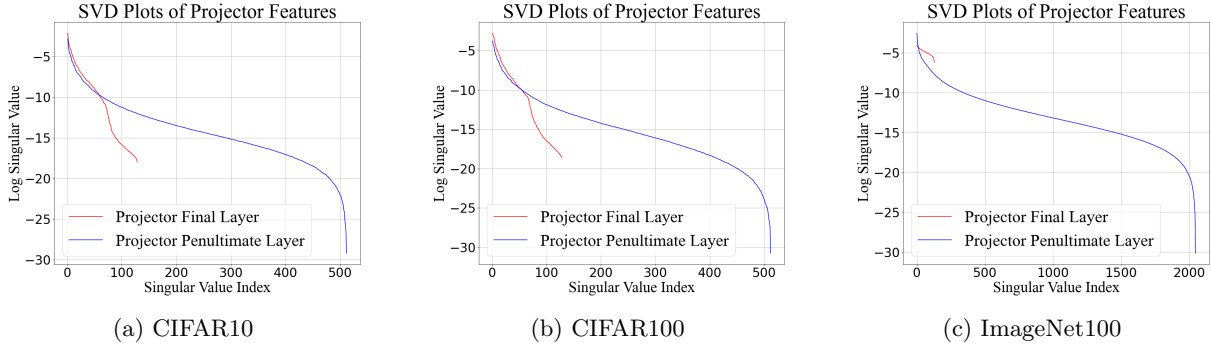

(a) CIFAR10          (b) CIFAR100          (c) ImageNet100

Figure 6: Singular values plots of the penultimate and final layers of the projector. Best viewed at 300%.

**Why are full-rank embeddings better than low-rank embeddings?** The trick used in DirectCLR is to leave a part of the output vector $z$ randomly initialized, theoretically making the variance of those dimensions non-zero. This causes the encoder to learn representations that are effectively higher-dimensional. Therefore, separability is also better according to *Cover's theorem* (Cover, 1965). Whereas, when the rank gets reduced, the representations are mapped to a low-dimensional subspace which *should* reduce the probability that the mapped instances are linearly separable while still being embedded in a high-dimensional space.

A similar approach based on Cover's theorem has also been taken in RankMe (Garrido et al., 2023) for the realisation of their framework.

**Proposition 2:** Low-rank embeddings from shallower layers are **not solely** responsible for dimensional collapse despite the decorrelating effects of InfoNCE loss.

**Proof:** Assuming that the norm of the weight matrices in the last layer is not zero, to investigate the reason for dimensional collapse, we need to look into the shallower layers and investigate the characteristics of the layer weights to prove Proposition 2. Assuming $W_l$ and $W_{l-1}$ are the weight matrices of two consecutive

layers in a linear network without skip connection or activation, $W_l^i$ is the $i$-th row of the weight matrix of the $l$-th layer.

Taking $x_l$ and $x_{l-1}$ are the inputs to the $l$-th and $(l-1)$-th layers, following Eqn. 4, we can directly get the result,

$$h_i = W_l^i x_l^T = W_l^i . \left( W_{l-1} x_{l-1}^T \right) = \left( W_l^i W_{l-1} \right) x_{l-1}^T \tag{9}$$

$$
\begin{aligned}
Var(h_i) =& W_l^i \left( x_l^T - \mathbb{E}[x_l^T] \right) \left( x_l^T - \mathbb{E}[x_l^T] \right)^T \left( W_l^i \right)^T \\
=& W_l^i . \left( W_{l-1} . x_{l-1}^T - \mathbb{E}[W_{l-1} . x_{l-1}^T] \right) \left( W_{l-1} . x_{l-1}^T - \mathbb{E}[W_{l-1} . x_{l-1}^T] \right)^T . (W_l^i)^T \\
=& (W_l^i . W_{l-1}) . \left( x_{l-1}^T - \mathbb{E}[x_{l-1}^T] \right) \left( x_{l-1}^T - \mathbb{E}[x_{l-1}^T] \right)^T . (W_l^i . W_{l-1})^T \\
=& (W_l^i . W_{l-1}) \Sigma_{l-1} (W_l^i . W_{l-1})^T \\
=& \| (W_l^i . W_{l-1}) \Sigma_{l-1}^{\frac{1}{2}} \|_2^2
\end{aligned}
\tag{10}
$$

where $\Sigma_{l-1} = \left( x_{l-1}^T - \mathbb{E}[x_{l-1}^T] \right) \left( x_{l-1}^T - \mathbb{E}[x_{l-1}^T] \right)^T$

Now, if $Var(h_i)$ for collapsed dimension of the output embedding $h$ becomes zero or very close to zero, then we can consider $Var(h_i) < \epsilon$, where $\epsilon > 0$ is very small.

$$Var(h_i) = (W_l^i . W_{l-1}) \Sigma_{l-1} (W_l^i . W_{l-1})^T < \epsilon \implies \left( W_l^i . W_{l-1} \right) \Sigma_{l-1}^{\frac{1}{2}} < \sqrt{\epsilon} = \epsilon' \tag{11}$$

**Case 1:** Now, $\Sigma_{l-1}$ is positive semi-definite, hence $\left( W_l^i W_{l-1} \right)$ lies in the approximate left nullspace (Ito et al., 2010) $\mathcal{N}_0(\Sigma_{l-1}; \epsilon) = \{ w_i : \lambda_i < \epsilon \}$ of $\Sigma_{l-1}^{\frac{1}{2}}$, that is,

$$W_l^i W_{l-1} \in \mathcal{N}_L(\Sigma_{l-1}; \epsilon) \tag{12}$$

where $\lambda_i$ are the eigenvalues of $\Sigma_{l-1}$, and a few of them are small but not exactly zero. This approximate nullspace interpretation explains why dimensional collapse in empirical networks is gradual rather than discrete, corresponding to near-zero singular values of the embedding covariance matrix. Hence, each collapsed embedding corresponds to a row of the last-layer weight matrix that lies approximately in the left nullspace of $\Sigma_{l-1}^{\frac{1}{2}}$.

**Case 2:** Another case can also occur, where the $\Sigma_{l-1}$ does not have small eigenvalues.

If $\lambda_{min}$ and $\lambda_{max}$ are the minimum and maximum eignevalues of $\Sigma_{l-1}$, then,

$$\lambda_{min} \| W_l^i W_{l-1} \|^2 \leq Var(h_i) \leq \lambda_{max} \| W_l^i W_{l-1} \|^2 \tag{13}$$

following Rayleigh quotient bound. So, if $Var(h_i) < \epsilon$, then

$$\| W_l^i W_{l-1} \| \leq \sqrt{\frac{\epsilon}{\lambda_{min}}} = \epsilon_l \tag{14}$$

Hence, we consider the case where $W_l^i$ lies in the approximate left null space of $W_{l-1}$, that is, $W_l^i \in \mathcal{N}_0(W_{l-1}, \epsilon_l)$.

We can have two cases here.

1. $\| W_l^i \|$ is small.

2. $W_l^i$ aligns with directions of $W_{l-1}$ with small eigenvalues (approximate left nullspace), indicating collapsed weight dimensions.

3. Combination of both small $\|W_l^i\|$ and alignment with eigen-directions with small eigenvalues.

Now, a dimension-collapsed representation from a previous layer will also diminish the covariance of the next layer's output embedding. From Eqn. 7,

$$Cov(h_i, h_k) = \left(x_l^T - \mathbb{E}[x_l^T]\right)^T . W_l^{iT} W_l^k . \left(x_l^T - \mathbb{E}[x_l^T]\right) \tag{15}$$

**Key Takeaway 2: Dimensional collapse does not necessarily result from low-rank Embeddings propagated from shallower layers** A low-variance or near-collapsed embedding dimension at layer $l$ $(Var(h_i) \leq \epsilon)$ can arise from multiple, non-exclusive factors. First, if the input covariance $\Sigma_{l-1}$ has small or zero eigenvalues along certain directions, the corresponding output variance is necessarily small. Second, even when $\Sigma_{l-1}$ is full-rank, a small composite norm $\|W_l^i W_{l-1}\|$ can produce low variance; this can occur due to small last-layer row norms, alignment with low-energy directions (approximate nullspace) of $W_{l-1}$, or a combination thereof. Therefore, low-rank or collapsed embeddings at layer $l$ do not automatically imply low-rank weight matrices in shallower layers — the collapse may result from input degeneracy, composite alignment, or row weakening in the last layer. This interpretation differs from WeRank (Pasand et al., 2024), which attributes collapse mainly to low-rank weights in earlier layers. Here, we provide a mathematical explanation showing that the observed collapse arises from the interaction between input covariance structure and composite layer transformations, rather than being a direct consequence of shallow-layer rank deficiency.

### 3.6.3 Does stopping information flow along some dimensions of the encoder output result in a better information bottleneck than using a Projector?

We attempt to substantiate our aforementioned statements more concretely and propose a straightforward approach to enhance performance on self-supervised contrastive learning tasks without requiring a projector. According to Property 5 described in Fang et al. (2024), a subvector of fixed value is the same as dimensional collapse along the dimensions of the subvector. We intend to stop the information flow, resulting in an enforced dimensional collapse to cause an information bottleneck without the projector by fixing the output of the embeddings along a few dimensions to a constant $k$. First, let us go through the notations. We only keep $d_0$ out of a total of $D$ dimensions as *dynamic*, while making the rest $(D - d_0 = d_r)$ *static* by assigning a constant output value to those dimensions of the embedding. Making the $i$-th dimension of the encoder output $h$, that is, $h_i$ static with a constant $k = 0$, stops the gradient flow through all paths connected directly or indirectly to $h_i$, and the rank of the covariance matrix $\mathcal{C}$ follows Eqn. 16. However, the weights $W^i$, which result in $h_i$, are still randomly initialized. Whereas, in DirectCLR, due to the randomized subvector, the rank of the covariance matrix does not collapse drastically (Eqn. 17).

$$\text{rank}(\mathcal{C}) \leq d_0 \tag{16} \qquad\qquad d_r \leq \text{rank}(\mathcal{C}) \leq d_r + d_0 \tag{17}$$

Thus, when a constant value is not assigned to the *static* $d_r$ dimensions, the rank of the encoder output embedding $h$ or consequently, the random variable $\mathcal{H}$ is less than when the *static* $d_r$ dimensions are left untouched.

Increasing the value of $d_0$ reduces the explicit dimensional collapse enforced on the representation space by allowing the rank of the embedding covariance matrix to increase according to Eqn. 16. Let us denote the new value of $d_0$ and $d_r$ be indicated by $d_0'$ and $d_r'$, where $d_0' > d_0$ and $d_r' < d_r$. The rank of the new embedding covariance matrix also increases. So, we may think that we are effectively increasing the shattering capacity by (a) increasing the rank of the embedding covariance matrix, (b) decreasing the degree of dimensional collapse (Fang et al., 2024), and also (c) increasing the dimensionality of the representation learning subspace (Cover, 1965). However, it is not the case as observed from Table 2. The gradient of the InfoNCE loss $\mathcal{L}$ with respect to an embedding $z_i$, is given by the Eqn. 18.

$$\frac{\partial \mathcal{L}}{\partial z_i} = \left[ -\frac{z_{i+}}{\tau} + \frac{\frac{z_{i+}}{\tau} \cdot e^{s_{ii+}} + \sum_{\substack{j=1 \\ j \neq i}}^{N} \frac{z_j}{\tau} \cdot e^{s_{ij}}}{e^{s_{ii+}} + \sum_{\substack{j=1 \\ j \neq i}}^{N} e^{s_{ij}}} + \sum_{\substack{j=1 \\ j \neq i}}^{N} \frac{\frac{z_j}{\tau} \cdot e^{s_{ji}}}{e^{s_{jj+}} + \sum_{\substack{k=1 \\ k \neq j}}^{N} e^{s_{jk}}} \right]$$

$$= - \left[ \frac{z_{i+}}{\tau} \left( 1 - p^{i \Downarrow i^+} \right) - \sum_{\substack{j=1 \\ j \neq i}}^{N} \frac{z_j}{\tau} \left( p^{i \Downarrow j} + p^{j \Downarrow i} \right) \right] \tag{18}$$

The quantity $p^{i \Downarrow j}$ is the probability of the pair $(x_i, x_j)$ being predicted as a positive pair with the sample $x_i$ as the anchor. The gradient along each dimension can be written as follows,

$$\frac{\partial \mathcal{L}}{\partial z_i^d} = \begin{cases} -\frac{k}{\tau} \left[ \left( 1 - p^{ii^+} \right) - \sum_{\substack{j=1 \\ j \neq i}}^{N} \left( p^{i \Downarrow j} + p^{j \Downarrow i} \right) \right] \\ \qquad \text{for } d_0 < d \leq d_0 + d_r \\ - \left[ \frac{z_{i+d}}{\tau} \left( 1 - p^{ii^+} \right) - \sum_{\substack{j=1 \\ j \neq i}}^{N} \frac{z_j^d}{\tau} \left( p^{i \Downarrow j} + p^{j \Downarrow i} \right) \right] \\ \qquad \text{for } d \leq d_0 \end{cases} \tag{19}$$

Therefore, $\frac{\partial \mathcal{L}}{\partial z_i^d} = 0$ if $k = 0$ for the subvector $h[d_0 : d_0 + d_r]$, whereas the gradient flows normally through the rest of the dimensions. Using a constant other than 0, causes a small gradient to flow through all $d_r$ dimensions. This gradient disrupts proper training of the kernel weights, since the gradient of the $d_r$ subvector $(h[d_0 : d_0 + d_r])$ points toward $\mathbb{1}_{d_r}$, which will eventually lead to dimensional collapse or may lead all points to lie within an open ball of finite radius along each of the $d_r$ dimensions. The performance, in this case, is worse than training with zero value in the $d_r$ dimensions and did not contribute towards maximizing the flow of information through the $d_0$ dimensions, as in DirectCLR or our framework with $k = 0$, due to the injection of a non-converging gradient. Empirical results for the CIFAR datasets are provided in Tables 2.

Table 2: 200-NN Accuracy for different values of $d_0$ and $d_r$ on CIFAR10 and CIFAR100 dataset.

| $d_0$ | $d_r$ | Fixed Value | CIFAR10 | CIFAR100 |
|---|---|---|---|---|
| | | | 200-NN Acc. | |
| 200 | 312 | 0 | 83.20 | 49.4 |
| 200 | 312 | $\frac{1}{\sqrt{512}}$ | 82.5 | 48.6 |
| 200 | 312 | 1 | 78.9 | 41.1 |
| 400 | 112 | 0 | 84.2 | 52.2 |
| 400 | 112 | $\frac{1}{\sqrt{512}}$ | 84.0 | 50.9 |
| 400 | 112 | 1 | 79.8 | 44.3 |
| 480 | 32 | 0 | 84.7 | 52.5 |
| 480 | 32 | $\frac{1}{\sqrt{512}}$ | 84.4 | 51.8 |
| 480 | 32 | 1 | 80.9 | 45.9 |

**Key Takeaway 3: Forced Collapse does not enforce an Information Bottleneck** The empirical results provided in Tables 2 combined with Fig. 7 indicate that we cannot observe the same information bottleneck effect without a projector. From the discussion in this subsection, we can safely say that the role of the projector is not only that of an information bottleneck, which is observed from the ineffectiveness of the forced collapse on the encoder output. The driving factor behind the effectiveness of the projector is that it allows the encoder to learn more high-level representations and, consequently, better separability with a higher rank of the embedding covariance matrix. Whereas a collapse in the last layer of the encoder prevents it from learning useful representations, which are essential for the effective classification of the input samples, as the norm of several kernels will be reduced to zero. With a constant subvector in the output, the weights

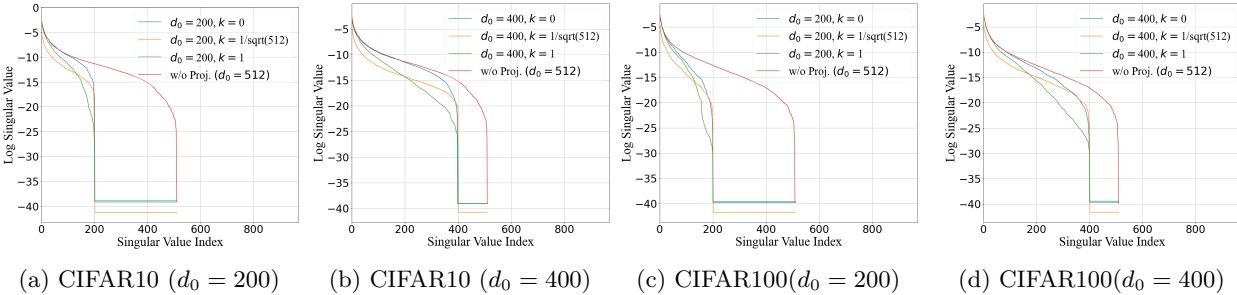

(a) CIFAR10 ($d_0 = 200$)  (b) CIFAR10 ($d_0 = 400$)  (c) CIFAR100($d_0 = 200$)  (d) CIFAR100($d_0 = 400$)

Figure 7: Singular values plots of encoder outputs embeddings when using a constant subvector in the output embeddings for different values of $d_0$ on CIFAR10 and CIFAR100 datasets. The above figures show the effect on the singular values of the encoder output embeddings, and it is evident that a forced collapse does not have the same effect as a natural dimensional collapse . Best viewed at 300%.

are not updated to learn useful representations. Furthermore, being unable to learn essential representations also diminishes the mutual information between the input and the output and the generalization error bound, which we prove in the next subsection.

### 3.7 Proposed Method: Remedy based on the Takeaways

For all the above key takeaways, we devise a single solution. We apply a *weight regularization loss similar to* WeRank *but only on the last layer*. Applying this weight regularization in the last layer prevents the norm of the weights from dropping to zero and thus prevents a significant drop in the variance of the output embeddings along the collapsed dimensions. Without loss of generality, applying induction logic from Eqn. 11, we can say that these high-rank embeddings prevent the collapse of representations in the previous layers. Thus, applying weight regularization remedies the low-rank embedding propagation. Lastly, an information bottleneck is ensured by maximizing the mutual information between the input and the output. Preventing the weight norm from dropping to zero maximizes the learning capacity of the network, thereby increasing the information flow between the input and the output.

**Proposition 3 :** A regularized weight matrix norm facilitates more effective mutual information maximization.

**Proof:** Let us consider the last $d$ dimensions to be susceptible to collapse, such that $\|W_i\| < \epsilon$ for $i \in [D - d + 1, D]$ and $\epsilon$ is arbitrarily small. The mutual information between two random vectors $z_k = \{z_k^1, z_k^2, \ldots, z_k^D\}$ and $z_l = \{z_l^1, z_l^2, \ldots, z_l^D\}$ forming a positive pair, can be expressed as,

$$
\begin{aligned}
\mathcal{I}(z_k, z_l) &= H(z_k) - H(z_k|z_l) \\
&= \sum_{d=1}^{D} H(z_k^d|z_k^{<d}) - \sum_{d=1}^{D} H(z_k^d|z_k^{<d}, z_l) \\
&= \sum_{d=1}^{D} \left[ H(z_k^d|z_k^{<d}) - H(z_k^d|z_k^{<d}, z_l) \right] = \sum_{d=1}^{D} \mathcal{I}(z_k^d; z_l|z_k^{<d})
\end{aligned}
\tag{20}
$$

Now, we can do the same decomposition over $z_l$, that is,

$$
\mathcal{I}(z_k, z_l) = \sum_{d=1}^{D} \left[ H(z_l^d|z_l^{<d}) - H(z_l^d|z_l^{<d}, z_k) \right] = \sum_{d=1}^{D} \mathcal{I}(z_l^d; z_k|z_l^{<d})
\tag{21}
$$

Combining both Eqn. 20 and 21, we can get an,

$$
I_{dd'} = \mathcal{I}(z_k^d; z_l^{d'}|z_k^{<d}, z_l^{<d'})
\tag{22}
$$

such that,

$$\mathcal{I}(z_k; z_l) = \sum_{d=1}^{D} \sum_{d'=1}^{D} \mathcal{I}(z_k^d; z_l^{d'} | z_k^{<d}, z_l^{<d'}) \tag{23}$$

Now, if some dimensions of $z_k$ or $z_l$ are collapsed, then the corresponding conditional mutual information will vanish. Thus, the total mutual information reduces to contributions only from the non-collapsed dimensions.

**Why does the collapsed dimension decrease mutual information?** If the collapsed dimensions have small but non-zero variance $(Var(h_i) = \sigma_i^2 < \epsilon$, without loss of generality, we can consider those dimensions to have a constant value $c$, resulting in $z_k^d$ being a deterministic function of $z_k^{<d}$. In that case,

$$H(z_k^d | z_k^{<d}) = 0 \text{ and } H(z_k^d | z_k^{<d}, z_l) = 0 \tag{24}$$

Hence,

$$\mathcal{I}(z_k^d; z_l | z_k^{<d}) = 0 \tag{25}$$

Putting Eqn. 25 in Eqn. 23, we get,

$$\mathcal{I}(z_k; z_l) = \sum_{d,d' \in \text{non-collapsed}} \mathcal{I}(z_k^d; z_l^{d'} | z_k^{<d}, z_l^{<d'}) \tag{26}$$

Considering the above deductions, we intend to apply weight regularization in the weight matrix of the last layer, which, by maintaining full-rank covariance, the encoder preserves information across all dimensions, thereby maximizing mutual information between embeddings.

In that case, $\mathcal{I}(z_k, z_l)_{collapsed} < \mathcal{I}(z_k, z_l)_{W_i W_i^T = I}$.

Hence, we can safely say that a non-negligible weight matrix norm facilitates better mutual information maximization.

To prove that our interpretation of the role of the projector in self-supervised contrastive learning is correct, we regularize the weight matrix $W$ of the last layer only by minimizing the regularization loss $\mathcal{L}_{reg} = \|WW^T - I\|^2$, in addition to the conservative loss in Eqn. 1. Thus, the final loss is described as,

$$\mathcal{L}_{total} = \mathcal{L}_{infonce} + \alpha \mathcal{L}_{reg} \tag{27}$$

Based on the above objective function, we optimize the parameters of the neural network following the implementation details outlined in Sec. 5. The results obtained are presented in the section below. We present the results on the datasets CIFAR10, CIFAR100, and ImageNet100 and also present eigenvalue plots for the CIFAR datasets to compare the effect of the proposed regularization towards the prevention of dimensional collapse. For all our experiments, we use $\alpha = 0.1$ for optimal performance following WeRank.

Table 3: Comparison of results obtained by applying WeRank variations to SimCLR without Projector, and our proposed strategy on ImageNet100 datasets. Here, 'LL' refers to 'Last Layer'. $(+/- \cdot)$: change from the previous model variation. We can observe that the proposed methods outperform the baseline SimCLR (vanilla) and WeRank on almost all cases (except one). This proves the effectiveness of the proposed regularization strategy on preventing degradation of performance due to dimensional collapse.

| Method | ImageNet100 |
|---|---|
| SimCLR (vanilla) w/o Projector | 45.82 |
| SimCLR w/o Proj. + WeRank (Full Enc.) | 44.68 (-1.14) |
| SimCLR w/o Proj. + Wt. Reg. LL (Ours) | **46.82** (+1.0) |

# 4 Results and Analyses

## 4.1 Comparison with state-of-the-art Contrastive learning frameworks

In this subsection, we analyse the efficacy of the proposed solution on different self-supervised frameworks, both with and without a projector. From Table 4, we can observe that the proposed solution successfully improves the kNN accuracy of all the SSL frameworks used and also reduces the dimensional collapse issue due to the low variance of feature dimensions in Fig. 8 (in Section 4.2). We also conduct experiments on the ImageNet100 dataset, and compare our proposed strategy with vanilla SimCLR and SimCLR added with WeRank, all trianed without a projector. We observe from Table ?? that the proposed strategy outperforms the two baselines on ImageNet100.

Table 4: Comparison of results obtained by applying WeRank variations to SimCLR with and without Projector, DirectCLR and our proposed strategy on CIFAR10 and CIFAR100 datasets. Here, 'LL' refers to 'Last Layer'. 'CS' refers to 'Constant Subvector'. (+/- ·): change from the previous model variation. We can observe that the proposed methods outperform the baseline SimCLR (vanilla) and WeRank on almost all cases (except one). This proves the effectiveness of the proposed regularization strategy on preventing degradation of performance due to dimensional collapse.

| Method | CIFAR10 | CIFAR100 |
|---|---|---|
| SimCLR (vanilla) | 86.1 | 56.3 |
| SimCLR + WeRank (Pasand et al., 2024) | **86.5** (+0.4) | 56.8 (+0.5) |
| SimCLR + Wt. Reg. LL (Ours) | 86.3 (+0.2) | **57.5** (+1.2) |
| SimCLR (vanilla) w/o Projector | 84.5 | 52.8 |
| SimCLR w/o Proj. + WeRank (Full Enc.) | 84.9 (+0.4) | 52.6 (-0.2) |
| SimCLR w/o Proj. + Wt. Reg. LL (Ours) | **85.1** (+0.6) | **53.1** (+0.3) |
| DirectCLR (vanilla) | *85.2* | **53.2** |
| DirectCLR+ WeRank (Full Enc.) | 85.4 (+0.2) | 53.0 (-0.2) |
| DirectCLR+ Wt. Reg. LL (Ours) | **85.5** (+0.3) | 53.2 (+0.0) |
| SimCLR w/o Proj (w/ CS) | 84.7 | 52.5 |
| SimCLR w/o Proj (w/ CS) + WeRank (Full Enc.) | 85.0 (+0.3) | 53.0 (+0.5) |
| SimCLR w/o Proj (w/ CS) + Wt. Reg. LL (Ours) | **85.5** (+0.8) | **53.1** (+0.6) |

## 4.2 Eigenvalue plots comparing SimCLR Encoder and our method

From Fig.8 we can see that when weight regularization is performed on the last layer of the encoder network, the singular values of the output embeddings improve across all dimensions. Thus, our method can reduce the effect of dimensional collapse due to the low variance of feature dimensions and provide better performance.

# 5 Implementation Details

**Datasets:** We primarily used three datasets for our study: CIFAR10, CIFAR100 (Krizhevsky, 2009), and ImageNet100 (Tian et al., 2020). CIFAR10 and CIFAR100 datasets consist of 10 and 100 classes, respectively, with 50K samples in the training set. ImageNet100 contains 1300 images in each of the 100 classes.

**Pre-training Details:** For experiments on CIFAR (Krizhevsky, 2009) and ImageNet, we used ResNet18 and ResNet50 as a backbone with the same modifications as done in SimCLR (Chen et al., 2020a) for small-scale datasets (CIFAR). We used a batch size of 128 and 256 for CIFAR and ImageNet, respectively. For the CIFAR and ImageNet datasets, we used SGD and LARS optimizer, respectively. All the implementations were done using *lightly-ai* (Susmelj et al., 2020) library. The value of the temperature hyper-parameter was set to 0.2 for all experiments. The value of $\alpha$ was set to 0.1 for all experiments, as using a higher value degrades performance. For the CIFAR dataset, the initial learning rate was set to 0.06 and decayed following

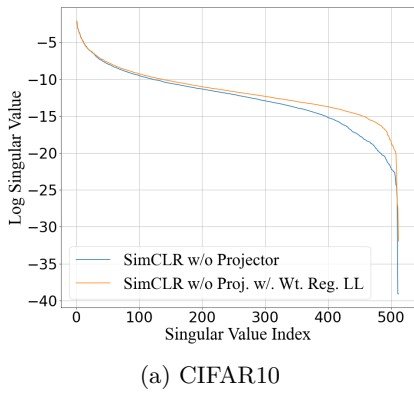
(a) CIFAR10

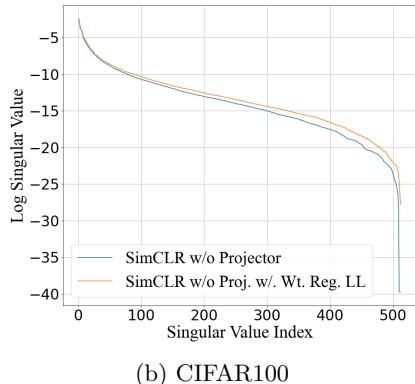
(b) CIFAR100

Figure 8: Singular values plots of encoder outputs, embeddings of SimCLR without a projector, and our method (last layer weight regularization) on CIFAR10 and CIFAR100 datasets. The plots clearly exhibit an improvement in the singular values, which indicates that the proposed regularization strategy prevents the dimensional collapse in the encoder output embeddings, when used without an additional projector.

a cosine schedule, whereas for the ImageNet dataset an initial learning rate of 0.3 was used and decayed following the same scehdule as the CIFAR datasets. For all the datasets, the training was conducted for 200 epochs only.

For the computation of the SVD decomposition, we simply used the *svd* function from the *numpy* library, following DirectCLR Jing et al. (2022).

## 6 Conclusion

In this work, we investigate the main reason behind the effectiveness of the projector in preventing dimensional collapse. We analyze mathematically the phenomenon that happens inside the projector and the encoder when trained without a projector. We find that the projector not only creates an information bottleneck but also facilitates the learning of high-level representations in the encoder, which does not occur without the projector, as a dimensional collapse occurring at the output of the encoder output prevents the learning of high-level representations. We also devise a solution to improve performance by only using a weight regularization in the last layer, be it with or without a projector, and achieve performance better than WeRank, which uses weight regularization over the whole network. We leave the study of the cause of dimensional collapse for our future work.

### Broader Impact Statement

This paper presents work whose goal is to advance the field of Machine Learning. There are many potential societal consequences of our work, none of which we feel must be specifically highlighted here.

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
