# OpenReview forum: "Can Weight Regularization in the Last Layer Reduce Dimensional Collapse?"
_TMLR — Rejected by TMLR_

### Review · Reviewer_5LgN · 2025-08-27

**Summary Of Contributions:**

*Summary*:
The projector has been found to heavily influence the quality of representations. This paper analyzed the role of the projector in self-supervised learning by examining the rank dynamics. Through empirical study and analysis, the phenomenon that happens inside the projector and the encoder when trained without a projector is analyzed. This paper also proposed a new strategy that modifies the WeRank method by applying weight regularization only in the last layer, for both with and without a projector.


*Strengths*:

1. This paper investigated the role of the projector in self-supervised contrastive learning by examining the rank dynamics through both empirical study and mathematical analysis.

2. The case in which the encoder without projector for contrastive self-supervised pre-training is also investigated.

3. This paper proposed a new strategy that applies weight regularization only in the last layer, and this strategy achieves better performance than the existing method that applies to the entire network.

*Weaknesses*:
1. This paper presented a detailed analysis to support its claims; however, the clarity could be improved. Summaries of the key takeaways and conclusions should be provided (see details in the Requested Changes).
2. The empirical experiment could include more details, particularly in the Implementation Details section.

**Additional Comments:**

1. Since the proposed method in Section 3.5 is based on Section 3.4, what is the relationship between Proposition 3 on page 10 and the other propositions and takeaways in the previous Section 3.4? Is Proposition 3 the specific theoretical support provided for the proposed method?

2. On page 8, ‘Detailed derivation in Appendix ??’ seems incomplete.

**Audience:**

Yes

**Audience Explanation:**

This paper builds on the findings of existing papers and summarizes the existing results while identifying gaps within these results. It analyzes the dynamics of the projector through rank decomposition and examines how the projector prevents dimensional collapse, which has not been investigated before. Additionally, this paper also proposes a modified method in Section 3.4, this method achieves better performance than the existing method, both with and without the projector.

**Broader Impact Concerns:**

No ethical concerns.

**Claims And Evidence:**

Yes

**Claims Explanation:**

The claims in this paper are supported by both empirical studies and analyses. The key conclusions and takeaways are supported by mathematical analyses and empirical analyses.
The proposed method in Section 3.5 is also supported by the mathematical analyses and the empirical verification in Sections 3.6 & 3.7.

**Requested Changes:**

1. In this paper, most of the results, takeaways, proposed method, and verification are all presented in Section 3, which makes the structure of Section 3 kind of unclear. Authors could consider reorganizing Section 3 into several sections. For example, the proposed method in Section 3.5, together with its verification in Sections 3.6 and 3.7, could be merged into a new section.

2. This paper provides a detailed, step-by-step analysis, leading to its claims. However, in the main text, the key conclusions are mixed with many detailed proofs. To make the main text cleaner and more concise, the authors could highlight the key conclusions and maintain a consistent format in the main results section (Sec 3), by adjusting the order of presentation or moving some of the proofs to the appendix.

3. For the empirical results presented in the tables and figures, their conclusions (takeaways) and some essential details could be included in the captions.

---

> ### Author Response · Authors · 2025-11-01
> **Reply to Reviewer 5LgN**
>
> > In this paper, most of the results, takeaways, proposed method, and verification are all presented in Section 3, which makes the structure of Section 3 kind of unclear. Authors could consider reorganizing Section 3 into several sections. For example, the proposed method in Section 3.5, together with its verification in Sections 3.6 and 3.7, could be merged into a new section.
>
> Thanks to the reviewer for this suggestion. We have subdivided the long section into several subsections for easier reading. However, the divisions slightly differ from the way suggested by the reviewer to manage the flow of the manuscript.
>
> > This paper provides a detailed, step-by-step analysis, leading to its claims. However, in the main text, the key conclusions are mixed with many detailed proofs. To make the main text cleaner and more concise, the authors could highlight the key conclusions and maintain a consistent format in the main results section (Sec 3), by adjusting the order of presentation or moving some of the proofs to the appendix.
>
> Thanks to the reviewer for this suggestion. As per your later suggestion, we have rectified the proof of Proposition 3 for better understanding. As suggested by the reviewer, we have included a necessary part of the proof in the appendix. However, due to the concerns of the other two reviewers regarding the other two propositions, we are keeping the proofs in the main body of the manuscript.
>
> > For the empirical results presented in the tables and figures, their conclusions (takeaways) and some essential details could be included in the captions.
>
> Thanks to the reviewer for the suggestion. We have made the necessary changes in the revised manuscript.
>
> > Since the proposed method in Section 3.5 is based on Section 3.4, what is the relationship between Proposition 3 on page 10 and the other propositions and takeaways in the previous Section 3.4? Is Proposition 3 the specific theoretical support provided for the proposed method?
>
> Thanks to the reviewer for this comment. We would like to highlight that Proposition 3 was meant to provide the necessary theoretical support for the proposed method. However, for better understanding, we have added more details in the proof of Proposition 3 (Page 13).
>
> > On page 8, ‘Detailed derivation in Appendix ??’ seems incomplete.
>
> Thanks to the reviewer for pointing this out. The mistakes have been taken care of in the revised manuscript.

---

### Review · Reviewer_ZcmN · 2025-08-28

**Summary Of Contributions:**

The paper proposes to study dimensional collapse in self-supervised learning methods, by trying to understand the role of the projector.
The authors produce a series of theoretical and empirical insights on existing SSL methods, such as how InfoNCE has a decorrelating effect, propositions surrounding the non-full rank of weight matrices in cases of collapse, further explanations on how DirectCLR avoids dimensional collapse, as well as how forced collapse does not necessarily correlate with information bottleneck.

Based on the analyses performed, the authors propose to regularize the weight matrix of the last layer of the network to help alleviate dimensional collapse. This is a more lightweight approach than previous works, yet demonstrates improved performance.

While the proposed method has practical benefits, some of the insights and theoretical results could be strengthened. This mostly comes down to clarity and presentation, as detailed in the rest of the review.

**Additional Comments:**

An appendix is mentioned on page 8, but I couldn’t find one with the submission.

Typo in page 2: “where the extra projector”, should probably be “where the extra predictor”.

**Audience:**

Yes

**Audience Explanation:**

Dimensional collapse is an issue in most, if not all, self-supervised learning methods, making solutions to it an important problem. The paper addresses this from both an understanding and practical point of view.

The proposed solution is easy to implement and not specific to a single method, making it widely applicable.

**Broader Impact Concerns:**

No specific concerns.

**Claims And Evidence:**

No

**Claims Explanation:**

The experimental results are convincing from a performance point of view, with gains across experiments. Looking at Figure 6 however, it is a bit unclear whether dimensional collapse is the biggest issue on these datasets, as we do not observe clearly collapsed representations as in DirectCLR or RankMe for example. Performance on ImageNet 100 (which shows more collapse) could help showcase the benefits of the method further.

Some of the presented results are contradictory in their current presentation. Prop 2: “Low rank embeddings from earlier are responsible…”, then Key Takeaway 2 ”dimensional collapse does not necessarily result from low-rank embeddings”. Prop 2 should be reformulated to make it clearer that low rank is a sufficient but not necessary condition.

Proposition 1 could be strengthened by looking at the weight matrices learned in practice to see whether or not rows become null. There is also an assumption that is not discussed which is that the embeddings $x$ must be non-collapsed before applying $W$. This is consistent with the general theme of the paper that dimensional collapse arises mainly at the last layer, but should be made clearer.

An assumption of Proposition 2 is that the network is linear, without skip connections or activation function. This should be clarified to make the relevance of the results clearer.

In general for both Proposition 1 and 2, collapsed dimensions are assumed to be aligned with the canonical basis of $R^N$, which may not always be the case. This assumption leads to clearer results, which is appreciated, but this simplification should be mentioned.

In Key Takeaway 1, the link between non-collapsed representations and learning “high level representations, which are more class-specific” is unsubstantiated, and seems like a strong jump from Proposition 1 which states that weights will be zero along collapsed dimensions. Further justification on how this is linked to not learning class-specific features would be needed.

Section 3.3 provides a similar analysis as [1], as shown in Figure S8 and appendix J for example. The results on ImageNet of [1] can help enrich the discussion here. Figure 2c and 2d are then used as proof that the output of the projector can be non-collapsed, even with a collapsed output of the encoder. However, as done throughout this work, an analysis of the singular values of the representations would be more relevant as it could help quantify the behaviour of the model beyond qualitative analysis.


[1] Garrido, Quentin, et al. "On the duality between contrastive and non-contrastive self-supervised learning." arXiv preprint arXiv:2206.02574 (2022).

**Requested Changes:**

See section “Are the claims made in the submission supported by accurate, convincing and clear evidence” for suggested improvements on the clarity and convincingness of the results. These are the most crucial points to address.

Presentation and clarity:
- Page 1, SimCLR is introduced as “using a large batch size” to avoid collapse. This misses the main point of using negative samples for contrastive learning, where large batch sizes are just a practical consideration and not the core anti-collapse mechanism.

- Page 2, “also uses a decorrelation loss in place of the hinge loss originally used in VICReg”. VICReg does use a decorrelation loss, and uses the hing loss to ensure that the variance is 1, akin to the normalisation done in Barlow Twins.

- Page 4 ”RankMe that InfoNCE loss fails to properly optimize [...] without a projector and results in collapse”. If I am not mistaken, all of the results in RankMe consider methods trained with a projector. The performance is then either evaluated on representations (encoder output) or embeddings (projector outputs). This should be clarified.

- Page 6, “in a concurrent work WeRank”. WeRank appeared more than a year ago, making the concurrency brittle. This should be omitted, as it does not remove any merit from the presented work.

- Page 6, “Why are full rank embeddings better”, the provided justification relying on Cover’s Theorem is very similar to what is used in RankMe[1], section 5. A mention of it here may be relevant

Missing references:
- The solution, as well as WeRank, are both similar to [2], which could be discussed to enrich the discussion about weight regularization in neural networks. Despite the name of this review section, this is not a “requested change” but a suggestion.

- The discussed self-supervised methods in the introduction and related work miss major developments in the field over the last few years, such as DINO [3], iBoT[4],DINOv2 [5], I-JEPA[6],  amongst others. I would encourage the authors to update the introduction and related works with more “recent” works.


[1] Garrido, Quentin, et al. "Rankme: Assessing the downstream performance of pretrained self-supervised representations by their rank." International conference on machine learning. PMLR, 2023.


[2] Zhu, Jiachen, et al. "Variance-covariance regularization improves representation learning." arXiv preprint arXiv:2306.13292 (2023).

[3] Caron, Mathilde, et al. "Emerging properties in self-supervised vision transformers." Proceedings of the IEEE/CVF international conference on computer vision. 2021.

[4] Zhou, Jinghao, et al. "ibot: Image bert pre-training with online tokenizer." arXiv preprint arXiv:2111.07832 (2021).

[5] Oquab, Maxime, et al. "Dinov2: Learning robust visual features without supervision." arXiv preprint arXiv:2304.07193 (2023).

[6]  Assran, Mahmoud, et al. "Self-supervised learning from images with a joint-embedding predictive architecture." Proceedings of the IEEE/CVF Conference on Computer Vision and Pattern Recognition. 2023.

---

> ### Author Response · Authors · 2025-11-01
> **Reply to Reviewer ZcmN**
>
> > The experimental results are convincing from a performance point of view, with gains across experiments. Looking at Figure 6 however, it is a bit unclear whether dimensional collapse is the biggest issue on these datasets, as we do not observe clearly collapsed representations as in DirectCLR or RankMe for example. Performance on ImageNet 100 (which shows more collapse) could help showcase the benefits of the method further.
>
> Thanks to the reviewer for pointing this out. We have added the results on ImageNet100 for SimCLR, SimCLR + WeRank and SimCLR + Weight Regularization in the last layer (Ours), all without the projector, in Table 3 in the revised manuscript. Our results show that the proposed regularisation method outperforms SimCLR without a projector.
>
> > Some of the presented results are contradictory in their current presentation. Prop 2: “Low rank embeddings from earlier are responsible…”, then Key Takeaway 2 ”dimensional collapse does not necessarily result from low-rank embeddings”. Prop 2 should be reformulated to make it clearer that low rank is a sufficient but not necessary condition.
>
> Thanks to the reviewers for this comment. We have changed the statement in the proposition and the proof to better support the key takeaway in our revised manuscript (Page 9).
>
> > Proposition 1 could be strengthened by looking at the weight matrices learned in practice to see whether or not rows become null. There is also an assumption that is not discussed which is that the embeddings x must be non-collapsed before applying W. This is consistent with the general theme of the paper that dimensional collapse arises mainly at the last layer, but should be made clearer.
>
> Thanks to the reviewer for this suggestion. We have added violin plots (Figures 3 and 4) for the weight norms and variances for all the convolutional layers of the last ResNet block. We observe that compared to the vanilla SimCLR pre-trained without a projector, the proposed weight regularization shows larger deviation from the number of near-zero norms and variance dimensions, which shows empirical evidence of Proposition 1.
>
> Furthermore, we have also added the statement as requested by the reviewer (Prop. 1, Page 6).
>
> > An assumption of Proposition 2 is that the network is linear, without skip connections or activation function. This should be clarified to make the relevance of the results clearer.
>
> We have added the suggested changes in the revised manuscript (Prop 2, Page 9).
>
> > In general for both Proposition 1 and 2, collapsed dimensions are assumed to be aligned with the canonical basis of RN, which may not always be the case. This assumption leads to clearer results, which is appreciated, but this simplification should be mentioned.
>
> We have added the suggested changes in the revised manuscript, wherever applicable.
>
> > In Key Takeaway 1, the link between non-collapsed representations and learning “high level representations, which are more class-specific” is unsubstantiated, and seems like a strong jump from Proposition 1 which states that weights will be zero along collapsed dimensions. Further justification on how this is linked to not learning class-specific features would be needed.
>
> Thanks to the reviewer for pointing this out. We have revised our discussion in the Key Takeaway 1 to better explain our statements (Pages 7-8).
>
> > Section 3.3 provides a similar analysis as [1], as shown in Figure S8 and appendix J for example. The results on ImageNet of [1] can help enrich the discussion here. Figure 2c and 2d are then used as proof that the output of the projector can be non-collapsed, even with a collapsed output of the encoder. However, as done throughout this work, an analysis of the singular values of the representations would be more relevant as it could help quantify the behaviour of the model beyond qualitative analysis.
>
> Thanks to the reviewer for this comment. Upon study of the mentioned reference paper, we found that our findings are similar to the findings in [1] in Figure 2 of our manuscript. We also observe the presence of off-diagonal noise in the Covariance plots in our experiments. The only difference between the figures in our manuscript and the provided reference [1] is that the authors in [1] provided the first 64 dimensions of the covariance and gram matrix, whereas we have provided all 2048 dimensions of the ResNet50 encoder for ImageNet100.
>
> [1] Garrido, Quentin, et al. "On the duality between contrastive and non-contrastive self-supervised learning." arXiv preprint arXiv:2206.02574 (2022).
>
> > Presentation and clarity
>
> Thanks to the reviewer for these comments on presentation and clarity. We have made the suggested changes in the appropriate places in the revised manuscript.

---

> ### Author Response · Authors · 2025-11-01
> **Reply to Reviewer ZcmN (Continued)**
>
> > The solution, as well as WeRank, are both similar to [2], which could be discussed to enrich the discussion about weight regularization in neural networks. Despite the name of this review section, this is not a “requested change” but a suggestion.
>
> Thanks to the reviewer for this suggestion. We have added a few lines about [2] for enriching the discussion about weight regularization.
>
> > The discussed self-supervised methods in the introduction and related work miss major developments in the field over the last few years, such as DINO [3], iBoT[4],DINOv2 [5], I-JEPA[6], amongst others. I would encourage the authors to update the introduction and related works with more “recent” works.
>
> Thanks to the reviewer for this suggestion. We have added the discussion in the revised manuscript.
>
> > Additional Comments: An appendix is mentioned on page 8, but I couldn’t find one with the submission.
> Typo in page 2: “where the extra projector”, should probably be “where the extra predictor”.
>
> Thanks to the reviewer for pointing this out. The mistakes have been taken care of in the revised manuscript.
>
> **References**:
>
> [1] Garrido, Quentin, et al. "Rankme: Assessing the downstream performance of pretrained self-supervised representations by their rank." International conference on machine learning. PMLR, 2023.
>
> [2] Zhu, Jiachen, et al. "Variance-covariance regularization improves representation learning." arXiv preprint arXiv:2306.13292 (2023).
>
> [3] Caron, Mathilde, et al. "Emerging properties in self-supervised vision transformers." Proceedings of the IEEE/CVF international conference on computer vision. 2021.
>
> [4] Zhou, Jinghao, et al. "ibot: Image bert pre-training with online tokenizer." arXiv preprint arXiv:2111.07832 (2021).
>
> [5] Oquab, Maxime, et al. "Dinov2: Learning robust visual features without supervision." arXiv preprint arXiv:2304.07193 (2023).
>
> [6] Assran, Mahmoud, et al. "Self-supervised learning from images with a joint-embedding predictive architecture." Proceedings of the IEEE/CVF Conference on Computer Vision and Pattern Recognition. 2023.

---

### Review · Reviewer_ajvU · 2025-10-18

**Summary Of Contributions:**

This work studies the dimensional collapse in contrastive self-supervised learning. The authors present a comprehensive discussion and analysis for the functionalities of the projector. They show that the near-zero norms in the last-layer leads to the dimensional collapse, and whitening-style weight regularization only to the last layer improves the representation learning. Experiments across SimCLR (with/without projector) and DirectCLR on CIFAR10/100 and ImageNet100 show improved singular value spectra and small accuracy improvements.

**Audience:**

Yes

**Audience Explanation:**

The problem studied in this work is interesting to the community.

**Claims And Evidence:**

No

**Claims Explanation:**

The logic, organization and the clarity of the current manuscript is poor and hard to follow. There is also a number of claims not well-supported. Please find the details in the sections below.

**Requested Changes:**

Presentation and clarity could be largely improved:
- The notations were not properly defined;
- The theoretical setup was not properly elaborated;
- The discussion in Sec 3.4 seems to be interesting, while not properly organized and hard to read. For example, Proposition 2 claims that " Low-rank embeddings from earlier layers are responsible for dimensional collapse despite decorrelating effects of InfoNCE loss ", while Takeaway 2 contradicts it.

In addition, the writing regarding the theoretical part and the discussion are not formal:
- Lots of critical concepts were not formally illustrated, for example, "high-level representations", "dimensional collapse", "information bottleneck". It's also confusing on their associations, as well as impacts to the downstream performance;
- Without a properly defined setup, it's hard to understand and assess the theoretical contributions such as Proposition 1;
- In the meantime, Propositions are not formal. For example, what exactly is the "earlier layers" in Proposition 2?


The evidence presented in the paper is not sufficient to support the claims:
- There is no direct evidence for Key Takeaway 1;
- The results from Table 2 do not seem to demonstrate the benefits of the regularization.

---

> ### Author Response · Authors · 2025-11-01
> **Reply to Reviewer ajvU**
>
> > The notations were not properly defined
>
> Thanks to the reviewer for pointing this out. We have added a notation table to the revised manuscript in Table 1 of Sec. 3.1 (Notations).
>
> > The theoretical setup was not properly elaborated
>
> Thanks to the reviewer for pointing this out. The theoretical framework primarily revolves around a black box neural network, which outputs embeddings, followed by two MLP layers represented by the weight matrices Wl-1 and Wl or W for the last layer. The term h represents the output of the final layer, while x represents the input to the weight matrices. Additionally, for simplicity, we consider no activation or batchnorm for the last two weight matrices.
>
> We have added the theoretical setup description in Sec 3.6.1 (Page 6)
>
> > The discussion in Sec 3.4 seems to be interesting, while not properly organized and hard to read. For example, Proposition 2 claims that " Low-rank embeddings from earlier layers are responsible for dimensional collapse despite decorrelating effects of InfoNCE loss ", while Takeaway 2 contradicts it.
>
> Thanks to the reviewers for this comment. We have changed the statement in the proposition and the proof to better support the key takeaway in our revised manuscript (Page 9).
>
>
> > Lots of critical concepts were not formally illustrated, for example, "high-level representations", "dimensional collapse", "information bottleneck". It's also confusing on their associations, as well as impacts to the downstream performance;
>
> Thanks to the reviewer for this suggestion.  We have added a subsection in the revised manuscript (Sec. 3.3) named “**Definitions**”, where we have provided the definitions for dimensional collapse, information bottleneck and high-level representations, which makes the understanding of the rest of the paper easier for the readers.
>
> To address the reviewer's concern regarding the association of the terms with downstream performance, we need to explain how each term is relevant to the downstream task. The principle of information bottleneck is utilised in the downstream task to discard irrelevant information while retaining useful information related to the downstream task. The high-level representations, that is, the task-specific representations in the deeper layers, are also relevant to the downstream task. Finally, the dimensional collapse, which can occur for both the self-supervised pre-training and supervised training stages, causes the learning to occur in a lower-dimensional subspace rather than in the high-dimensional embedding space. A larger utilisation of the learning subspace results in better performance in the downstream task.
>
> > Without a properly defined setup, it's hard to understand and assess the theoretical contributions such as Proposition 1
>
> Thanks to the reviewer for pointing out this mistake. We would like to point out that we have described the setup in the paragraph preceding Proposition 1 in the revised manuscript. Moreover, in the revised manuscript, we have also added information about the dimensionality of the terms involved in the proof for better understanding (Prop 1, Page 6).
>
> > In the meantime, Propositions are not formal. For example, what exactly is the "earlier layers" in Proposition 2?
>
> Thanks to the reviewer for pointing this out. By the term “earlier”, we meant the shallower layers in a neural network, and we have replaced all occurrences of the term “earlier” layers throughout the manuscript with “shallower” layers for better understanding and correctness.
>
> > There is no direct evidence for Key Takeaway 1;
>
> Thanks to the reviewer for this comment. We have added the violin plots for the weight norms and unbiased variance for the last layer convolutional layers in ResNet encoder for SimCLR and SimCLR without proposed weight regularization, both trained without a projector, in Figures 3 and 4 (page 7).
>
>
> > The results from Table 2 do not seem to demonstrate the benefits of the regularization.
>
> We thank the reviewer for this comment. The proposed weight regularization has improved the performance over the vanilla SimCLR setting for all the cases, and also outperformed WeRank in terms of mean performance improvement. Please note that Table 2 is not Table 4 in the revised manuscript.

---

> > ### Comment · Reviewer_ajvU · 2025-11-20
> >
> > Thank the authors for the explanation. As I can not find direct responses to my questions in the comment from the authors, I checked the revised paper. I still found several concerns:
> >
> > - In the revised manuscript, I still found that several critical concepts were not formally defined, but just illustrated conceptually. And the revision in the manuscript even includes the response to reviewers, which is not professional.
> >
> > - Meanwhile, even for "shallower layers", which layers do you exactly refer to?
> >
> > - In Table 4, the improvements seem to be marginal.

---

> > > ### Author Response · Authors · 2025-11-21
> > > **Reply to Reviewer ajvU**
> > >
> > > > In the revised manuscript, I still found that several critical concepts were not formally defined, but just illustrated conceptually. And the revision in the manuscript even includes the response to reviewers, which is not professional.
> > >
> > > Thanks to the reviewer for this comment. We have tried to define the terms mentioned by the reviewer in the previous comments. However, we will look into the manuscript and try to define several other critical concepts not defined.
> > >
> > > Meanwhile, we have rectified the mistake pointed out by the reviewer in the latest version of the manuscript.
> > >
> > > > Meanwhile, even for "shallower layers", which layers do you exactly refer to?
> > >
> > > "Shallower" layers often refer to the layers that are closer to the input [1, 2, 3], whereas the term "deeper" layers refer to the ones closer to the output.
> > >
> > > > In Table 4, the improvements seem to be marginal.
> > >
> > > We agree with the reviewer that the improvement seems to be marginal. However, our primary focus in this work has been to study the phenomenon of dimensional collapse in self-supervised contrastive learning and investigate the reasons behind it, which we have done through different mathematical derivations in our manuscript. Furthermore, the improvement in our method comes at a much lower cost than WeRank [4] and also without a projector.
> > >
> > > References:
> > >
> > > [1] Ian Goodfellow, Yoshua Bengio and Aaron Courville. Deep Learning. : MIT Press, 2016.
> > >
> > > [2] 	Chien-Yao Wang, I-Hau Yeh, Hong-Yuan Mark Liao: You Only Learn One Representation: Unified Network for Multiple Tasks. J. Inf. Sci. Eng. 40(1): 691-709 (2024)
> > >
> > > [3] Hrushikesh Mhaskar, Qianli Liao, and Tomaso Poggio. 2017. “When and Why Are Deep Networks Better Than Shallow Ones?”. Proceedings of the AAAI Conference on Artificial Intelligence 31 (1). https://doi.org/10.1609/aaai.v31i1.10913.
> > >
> > > [4] Ali Saheb Pasand, Reza Moravej, Mahdi Biparva, Ali Ghodsi: WERank: Towards Rank Degradation Prevention for Self-Supervised Learning Using Weight Regularization, 4th Workshop on Self-Supervised Learning: Theory and Practice, NeurIPS 2023.

---

> > > > ### Comment · Reviewer_ajvU · 2025-11-25
> > > >
> > > > Thank the authors for the follow-up clarification. However, I still find the current manuscript lacking formalization and clarity as a theoretical paper. For example,
> > > > ```
> > > > Proposition 1 : The norm of the weight vectors associated with the collapsed dimensions becomes zero.
> > > > ```
> > > >
> > > > ```
> > > > Proposition 2 : Low-rank embeddings from shallower layers are not solely responsible for dimensional
> > > > collapse despite the decorrelating effects of InfoNCE loss.
> > > > ```
> > > > Terms are all illustrated in conceptual forms. It's also unclear about the underlying theoretical model, training objective, and convergence. Given the limited empirical improvements, I am leaning towards a weak reject, but am not opposed to acceptance.

---

### Author Response · Authors · 2025-11-01
**Reply to the Reviewers and AE**

Thanks to all the reviewers for their comments and suggestions. We have updated the manuscript to our best and tried to adhere to all the suggestions from the reviewers. The changes to the manuscript have been marked in blue throughout the manuscript.

The changes are as follows:

- Changed Proposition 2 and Key Takeaway 2 as per the suggestions from Reviewers ajvU and ZcmN
- Added empirical evidence for Proposition 1
- Added results for ImageNet100 (Reviewer ZcmN)
- Added theoretical setup (reviewer ajvU)
- Added table for notations (Reviewer ajvU)
- Added a separate subsection for the definition of several terms used in the paper for better understanding (Reviewer ajvU).
- Added a few more papers in the literature review (Reviewer ZcmN)
- Subdivided the Section in the previous version into several smaller subsections for better understanding and easier flow (Reviewer 5LgN).
- Changed the captions of figures and tables to include conclusions and essential details (Reviewer 5LgN).
- Improved Proposition 3 for better understanding as per the suggestion from the Reviewer 5LgN

Please let us know your further comments to improve the manuscript.

---

### Decision · Action_Editor_VG1C · 2025-12-19

**Recommendation:** Reject

**Audience:**

Yes

**Audience Explanation:**

This work studies interesting topics. Contrastive self-supervised learning and dimensional collapse are highly relevant and interesting topics for the TMLR audience.

**Claims And Evidence:**

No

**Claims Explanation:**

This work investigates the phenomenon of dimensional collapse in contrastive self-supervised learning, with a particular focus on understadning the projector. The authors demonstrate that near-zero norms in the final layer lead to dimensional collapse, and that applying whitening-style weight regularization specifically to the last layer enhances representation learning. While the reviewers agree that the topic studied in this paper is interesting, it has also been pointed out by the reviewers that the revised version of the manuscript still contains vague and informal statements. For example, the authors revised Proposition 2 as "Low-rank embeddings from shallower layers are not solely responsible for dimensional collapse despite the decorrelating e!ects of InfoNCE loss." This proposition is not rigorous, and it is not clear what are the mathematical definitions of "responsible", or "not solely responsible". Given the presence of vague theoretical statements and discussions, the paper does not meet the standard required for publication.